# Fabrication of Graphene-Metal Transparent Conductive Nanocomposite Layers for Photoluminescence Enhancement

**DOI:** 10.3390/polym11061037

**Published:** 2019-06-11

**Authors:** Hongyong Huang, Zhiyou Guo, Sitong Feng, Huiqing Sun, Shunyu Yao, Xidu Wang, Dong Lu, Yaohua Zhang, Yuan Li

**Affiliations:** 1Institute of Semiconductor of Science and Technology, South China Normal University, Guangzhou 510631, China; hhy_2635102@163.com (H.H.); sunhq@scnu.edu.cn (H.S.); yaoshunyu@m.scnu.edu.cn (S.Y.); 18403433625@163.com (Y.Z.); liamly@m.scnu.edu.cn (Y.L.); 2Guangdong Engineering Technology Research Center of Optoelectronic Functional Materials and Devices, South China Normal University, Guangzhou 510631, China; 3Guangdong Polytechnic, Foshan 528000, China; 4School of chemical and Light Industry, Guangdong University of Technology, Guangzhou 510006, China; f13414042384@163.com; 5Guang Zhou OED Technologies, Inc., No. 19 Guangsheng Road, Nashadistict, Guangzhou 511458, China; xdwang@cyrontech.com; 6Guangzhou HKUST Fok Ying Tung Research Institute, Guangzhou 511458, China; maeld@ust.hk

**Keywords:** transparent conductive layer, graphene-metal nanocomposite, electromagnetically induced transparency, surface plasmon

## Abstract

In this work, the synthesis and characterization ofgraphene-metal nanocomposite, a transparent conductive layer, is examined. This transparent conductive layer is named graphene-Ag-graphene (GAG), which makes full use of the high electron mobility and high conductivity characteristics of graphene, while electromagnetically induced transparency (EIT) is induced by Ag nanoparticles (NPs). The nanocomposite preparation technique delivers three key parts including the transfer of the first layer graphene, spin coating of Ag NPs and transfer of the second layer of graphene. The GAG transparent conductive nanocomposite layer possess a sheet resistance of 16.3 ohm/sq and electron mobility of 14,729 cm^2^/(v s), which are superior to single-layer graphene or other transparent conductive layers. Moreover, the significant enhancement of photoluminescence can be ascribed to the coupling of the light emitters in multiple quantum wells with the surface plasmon Ag NPs and the EIT effect.

## 1. Introduction

Photoluminescence (PL) enhancement has attracted the attention of many researchers [1,2,3]. Their methods include changing the structure of quantum wells, using high transparency and high conductivity conductive layers and proper device structures [4,5,6]. The transparent conductive layer is conductive and transparent, and it is widely used in optoelectronic devices [7,8,9]. The main transparent conducting layers are indium tin oxide (ITO) and zinc oxide (ZnO) [10,11,12,13]. The sheet resistance of ITO and ZnO are 157.9 ohm/sq and 100 ohm/sq [10,11], and their conductivities are 150 S/cm and 300 S/cm [12,13]. The electron mobility is of paramountimportance in that it determines the transverse extension of the current [14,15]. However, their electron mobilities are 32 cm^2^/(v s) and 69 cm^2^/(v s), which are too small to limit the lateral diffusion of the current [16,17,18]. In recent years, there have been many studies of graphene as a transparent conductive layer, but it has been found that the sheet resistance of single- layer graphene is particularly high, and the transparency of multilayer graphene has been reduced [19,20,21].

The exploitation of surface plasmon (SP) coupling in light-emitting diodes (LEDs) is a promising approach to achieve the photoluminescence enhancement [22,23,24]. Further, plasmon-enhancement has been reported in multiple quantum wells coated with silver (Ag) films [25,26]. The fundamental resonance mode of Ag nanostructures is located at a yellow-red spectral range [27], which can usually induce a stronger localized surface plasmon (LSP) coupling effect when compared with a high-order resonance mode. Electromagnetically-induced transparency (EIT) refers to the formation of a narrow transparency window within a broad absorption profile upon the application of an indirect excitation and the quantum destructive interference between the two excitation channels in a three-level atomic system [28,29,30]. This phenomenon leads to a narrow transparent spectrum through an opaque medium accompanied with strong dispersion, hence produces the ultraslow group velocity and enhanced light-matter interactions [31,32].

In this work, SP of Ag nanoparticles (NPs) and EIT of graphene-based metamaterials were well researched by the finite-difference time-domain method (FDTD). Graphene-Ag nanocomposite transparent conductive layer would be a suitablechoice due to the electron mobility of graphene and the SP and EIT effect of Ag NPs. The sheet resistance of the graphene-Ag-graphene (GAG) transparent conductive layer, which is synthesized by the authors, is 16.3 ohm/sq, and its electron mobility is 14,729 cm^2^/(v s).

## 2. Experimental Section

### 2.1. The Growth of Graphene Sheet Layer

First, the rolled copper (Cu) foil was ultrasonically cleaned with absolute ethanol, and then placed in vacuum plasma cleaner for 180 s. After plasma cleaning, the copper foil was loaded on the mold, and placed in a chemical vapor deposition (CVD) furnace, evacuated, and continuously passed through gas hydrogen (H_2_) and argon (Ar). Then, a vacuum is applied to the growth of the H_2_ and methane (CH_4_). After growing 12 min, the flow rates of both H_2_ and CH_4_ were adjusted to 200 mL/min. After 5 min, vacuum pumping, H_2_ and Ar are introduced. The furnace was cooled down to room temperature naturally.

### 2.2. Transfer Graphene Layer to Target Substrate

The as-synthesized G/Cu sample was taken out from the furnace and placed on the spin coater with dropping several drops of polymethyl methacrylate (PMMA) solution on it. After spin coating, the sample was heated in an oven at 150 °C for 20 min. After that, oxygen plasma was introduced to remove the graphene on the back side of the copper foil. The PMMA/G/Cu sample was immersed to the Ferric chloride (FeCl_3_) solution for 10 h and then employed a glass slide to collect the PMMA/G composite which floated on the FeCl_3_ solution. The composite was washed by hydrochloric acid (HCl) aqueous solution and de-ionized (DI) water. Following this, the PMMA/G was transferred to the target substrate and dried naturally. The PMMA/G/target substrate was respectively soaked in both acetone (CH_3_COCH_3_) and ethyl acetate (CH_3_COOC_2_H_5_) solution for 1 h each, and dried in the air for further use. Finally, the graphene was placed on the silica (SiO_2_) substrate.

### 2.3. Preparation of GA (Graphene/Ag) Hybrid Sample 

The preparation of a GA (Graphene/Ag) hybrid sample needed to be carried out in an oxygen-free, ultra-clean environment. Further, 0.05 mL of the purchased Ag solution (0.1 mg/mL) was added into 50 mL DI water to prepare an Ag solution with a concentration of 0.1 μg/mL. In order to ensure that Ag NPs distributed evenly on the graphene layer, a plasma cleaning technique was applied to improve the hydrophilicity of graphene. Typically, a piece of graphene layer with the size of 1 cm × 1 cm was placed in PDC-002 plasma cleaning machine for 15 min with high power. Following this, 0.1 mL of as-prepared Ag solution was dropped on the treated samples and spun in four different modes for comparison (a. 1000 rpm, 30 s; b. 600 rpm, 30 s; c. 600 rpm, 10 s and d. 0 rpm).

### 2.4. Transfer Second Layer Graphene to GA

Transferring the second layer grapheme to the GA also needed to be carried out in an oxygen-free, ultra-clean environment. Using the graphene which was prepared as shown in Section 2.1, graphene stripping was prepared as shown in Section 2.2. Finally, the graphene was placed on the GA. 

### 2.5. Theoretical Model

The graphene layer is reasonably considered as a sheet of material modeled with complex surface conductivity. The surface conductivity can be retrieved by the Drude model under THz region [33]:(1)σintra(ω,EF,Γ,T)=je2KBTπℏ2(ω+jΓ)(EFKBT+2ln(e−EFKB+1))
where *e*, *ћ*, and *K_B_* are the universal constants representing the electron charge, Planck’s constant, and Boltzmann’s constant, respectively. Further, *ω* is the radian frequency. In addition, j, *E_F_* and Γ (Γ = *ћ*/*τ*, where *τ* is the electron relaxation time) are the current density, and the physical parameters of graphene layer accounting for Fermi energy and intrinsic losses, respectively.

Simultaneously, the dielectric constant of the graphene layer can be expressed as follow:(2)$$\varepsilon_{g}=1+j\frac{\sigma_{g}}{\omega \varepsilon_{0}\Delta }$$
where Δ is the thickness of the graphene layer and *ε*_0_ is the permittivity. In our calculations, the parameters were set as follows: T = 300 K, Γ = 0.4 meV, and the thickness of graphene layer is Δ = 0.34 nm [34,35,36]. By comparison to traditional metal film, the conductivity of the graphene layer can be tuned by varying its chemical potential via electrostatic biasing from Equation (1).

The penetration length of the SPPs, *d*, is defined by Equation (3), where *ε_m_* and *ε_d_* are relative permittivities of the metal and dielectric, respectively. Further, *ω* is the frequency of the SPPs, and *c* is the speed of light. The relative permittivity of Ag is depended on the wavelengths by neglecting the imaginary part in the complex relative permittivity of a metal [19].
(3)$$d=\frac{c}{\omega }\sqrt{\left(\frac{\varepsilon_{m}+\varepsilon_{d}}{-\varepsilon_{m}^{2}}\right)}$$

### 2.6. Simulation Method

In Figure 1, the problem geometry, an embedded radiating dipole, only two orientations including the x- and y- directions are considered, at a fixed depth d, in an InGaN quantum well coupling with the LSP modes induced on a surface Ag NP. The refractive index of GaN-based layers are set as 2.4, including *p*-GaN, InGaN/GaN QW layer and *n*-GaN [37], and experimental data are used for the dielectric constant of Ag [27], and the refractive index of the capping layer is n. The three-dimension simulations are carried out with the assistance of the commercial software, FDTD Solutions, which is based on the numerical algorithm of the finite-difference time-domain (FDTD) method. The perfect matched layer (PML) boundaries condition is set in ±*z* directions and periodic boundaries condition which determines the period of the Ag NPs is set in ±*x* and ±*y* directions. The size of simulation region is 900 nm × 900 nm × 1000 nm for *x*, *y* and *z* directions, respectively. A power monitor surrounding the radiating dipole is used to measure the radiated power of the dipole via calculating the net total power flowing out of the monitor box. Another power monitor surrounding the Ag NP is used to measure the absorbed power of the NP (P*_abs_*) determined by its size and material. The radiated and absorbed power are normalized and defined as the ratio of the radiated and absorbed power with the NP over that without the NP, respectively. 

### 2.7. Characterization

Electron mobility can be obtained according to Equations (4) and (5), where d is the thickness of the sample, B is the magnetic field strength, q is the electron charge, $V_{\mathrm{BD}}$ is the voltage across the BD of the GAG sample, and $I_{\mathrm{AC}}$ is the current of the AC of the GAG sample.
(4)μ=|RH|ρ
(5)$$\left|R_{H}\right|=\frac{d}{B}\frac{\Delta V_{\mathrm{BD}}}{I_{\mathrm{AC}}}$$

A HMS-5000 full-automatic temperature Holzer effect tester is based on the principle design, which can intuitively obtain the electron mobility of GAG. Before the test, the GAG sample was fixed to the center of the PCB board with double-sided tape, and the side of the graphene was facing up. Using a thin metal wire to weld the four apex angles of the sample to the joints of the four corners of the PCB circuit board. In order to ensure the accuracy of the experimental data, it should be noted that the solder joints should be as small as possible and must be soldered with special indium silver. The length of 4 wires should be short and be similar. The PCB board was inserted into the rack and the rack was closed. The power of the experimental instrument was turned on and the experiment started after it was fully warmed up. In the experiment, the input current, magnetic field strength, and film thickness D in the Input Value section on the upper left side of the experiment interface needs to be filled in, and then the instrument automatically starts the experiment and data calculation. The GAG transparent conductive layer consists of two layers of graphene sandwich silver particles, so it can be approximated to a film material with uniform thickness. The electrical conductivity can be obtained by the Vanderbilt method. It is also tested by HMS-5000 full-automatic temperature Holzer effect tester. The test must note that the four contact points, A, B, C, and D, are as small as possible (far less than the sample size) and that the four contact points must be at the edge of the film.

The absorption spectrum of the GAG transparent conductive layer and graphene-Ag (GA) were tested by UNICOUV-2600/A ultraviolet spectrophotometer. The test must align the center of the sample with the exit aperture to ensure that the sample is perpendicular to the optical path. It is necessary to test the contrast silicon dioxide substrate for reference before testing.

## 3. Results and Discussion

Under the same conditions, the material of the metal ball is different, and the effect of plasma excitation is different. With the assistanceof FDTD, the plasma excitation of four kinds of nano-metal particles, aluminum (Al), silver (Ag), gold (Au), and chromium (Cr) nanoparticles, were studied under the same conditions, i.e., 50 nm radius and 300–800 nm light wave range. As shown in Figure 2, the effective radiated power (ERP) (SiO_2_) line shows the normalized radiated power (RP) of the light wave on the interface between the SiO_2_ and the air when there areno metal NPs on the SiO_2_ surface. The Au NPs have the best normalized radiated power at 614 nm (RP = 1.44), which is the closest to the yellow band. However, its half-wave width is too narrow to reach the peak effect in the actual preparation. It is a good choice to use the Ag NPs, because its wavelength width is 150 nm when its RP is greater than 1.

From Figure 2, the radiated power of Ag NPs is 1.31, but its effective radiated power (ERP) is only 0.93. Asthe absorption of the light wave is affected by the size of the NPs, the ERP must be the difference between the normalized radiated power (RP) and the absorbed power (P*_abs_*), so ERP = RP − P*_abs_* [38,39,40]. The effective radiated powers of NPs with different radius in the visible band were researched. In the radius setting, seven kinds of radius had been studied, which were 50 nm, 60 nm, 70 nm, 80 nm, 90 nm, 100 nm, and 110 nm, respectively. Figure 3 shows that the highest plasma-induced excitation peak is induced by the Ag NPs whose radius is 100 nm. It is the reason that the radius of NPs is chosen to be 100 nm in experimental preparation.

The period of the Ag NPs is mainly taken into account from two aspects—the plasma-induced excitation angle and the absorption [19,41,42,43,44]. The plasma excitation effect and the absorption of nanoparticles vary with the incident angle of the light wave. The longer the perimeter and the smaller the angle, the worse is the plasma excitation effect. When the incident angle is larger, the effect of the plasma-induced excitation is stronger, but the more NPs in the unit area, the more light is absorbed. The optimal number of NPs per unit area can be set by rational optimization of the periodicity, therefore, it is necessary to compare their ERP. Taking into account that the Ag NPs radius is 100 nm, the ERP and absorption of the six periods are studied. The six periods which are a regular set of periods selected from more than a dozen research periods are 500 nm, 700 nm, 900 nm, 1100 nm, 1300 nm, and 1500 nm, respectively, and their ERP are shown in Figure 4. The four cases with a period of 900 nm or above are better in the red and yellow spectral range. However, 1500 nm is in a downward trend in the whole yellow band, so the period can be controlled at 900~1300 nm. This result provides more ways for the experimental preparation of GAG, such as spin-coating and printing with a certain concentration of the Ag NPs.

The surface morphology of the graphene layer was examined by SEM and showed in Figure 5a. Surface wrinkles reflect the surface of graphene, and the Raman spectrum further verifies the existence and quality of graphene, which are showed in Figure 5b. The graphene has two Raman characteristics, which are G and 2D. The G belt is located at 1582 cm^−1^ and the 2D belt is located at 2700 cm^−1^. Asthe 2D band is generated by two phonon double resonance processes, it is related to the band structure of the graphene layer. The G band is produced due to the A-mode of the double-degenerate Brillouin zone center, which is also considered to be due to the presence of *sp*^2^ carbon (C). The Raman spectra of high quality single layer graphene show that the intensity of 2D is much greater than that of G. In multilayer graphene, 2D and G are exactly the opposite of single-layer graphene. The Raman peaks of as-prepared graphene layer were consistent with the reported data at [45,46,47], which implied the high quality of the CVD growth graphene, as illustrated in Figure 5b. The I-V curve of the graphene in Figure 6 also illustrated that the quality of the CVD growth graphene is very high because there was no obvious current fluctuation in different voltage measurements [48,49,50,51]. This depends on high precision experimental equipment and optimized preparation process.

Four spin-coated samples named GA samples were tested by JSM-6390 SEM after air-drying as shown in Figure 7. The 1000 r/min speed of sample A was so fast that most of the NPs were thrown out of the sample. The distances between Ag NPs were very large. The rotation time of sample B was too long, which caused the accumulation of Ag NPs in the periphery of the sample. Sample D was not rotated and dried directly, so the overall accumulation phenomenon was more serious. Sample C had moderate rotational speed and appropriate time. The Ag NPs distributed uniformly in centripetal force, and the edge also accumulated slightly due to the edge effect. It was obvious that the distribution of the Ag NPs in sample C was the most uniform.

Sample C was selected as the second graphene transfer sample, and the transfer process was the same as the first one. Due to the transparency of graphene, it could not be seen with SEM, so anatomic force microscope (AFM) was used to characterize the surface morphology of GAG, which isshown in Figure 8. The step line of GAG selected in a transverse region of the AFM was zoomed in, and indicated its difference from the step line of GG, which was very smooth (the test result is tilted due to the slightly tilted test base). It indicated that the fluctuation of the GAG layer was due to the Ag NPs.

The carrier motilities of graphene and the GAG transparent conductive layer were achieved, which were 11,524 cm^2^/(v s) and 14,729 cm^2^/(v s). The mobility of graphene is lower than that of high-quality graphene, but the mobility of GAG is close to the mobility of high-quality graphene. The sheet resistances of graphene and the GAG transparent conductive layer were 864 ohm/sq and 16.3 ohm/sq. As described earlier, the sheet resistance of single-layer graphene is still as large as that of [19,20,21]. After forming the GAG structure, the square resistance of the transparent conductive layer is already smaller than that of ITO and ZnO in [10] and [11]. ITO, GAG and GA were used as transparent conductive layers for the same type of LED, and their PL spectra shown in Figure 9 indicate that GA and GAG are more suitable for transparent conductive layers than ITO. The PL peak value of GAG is 15.625% higher than GA. The absorption spectrum of the GAG transparent conductive layer and graphene-Ag (GA) are basically consistent with the effective radiated power of GA and GAG by FDTD simulation.

The absorption spectrum of GA and GAG showed that the second graphene layer increases the GAG absorption. However, the GAG transparent conductive layer would increase the optic power due to plasma-induced enhancement which can be seen in Figure 10. The ERP of GAG is 1.33, and the ERP of GA is 1.27, therefore, the second layer of graphene increased the ERP by 4.7%.

A question to consider is why the effective radiated power of GAG is better than GA. If the simple transparency (T) formula ($T=T_{1}\times T_{2}\times T_{3}$), T_GAG_ must be $0.975T_{\mathrm{GA}}$. In this study, the spacing between two layers of graphene is 200 nm, so EIT is easily produced. Using FDTD, the current induced on GAG by 600 nm light waves is higher than GA (J_GAG_ (MAX) = 7 × 10^5^, J_GA_ (MAX) = 5.9 × 10^5^) (Figure 11). The electromagnetic induction effect also makes its carrier motilities higher than grapheme [51,52,53]. The thickness of GAG is 590 times that of graphene, and the electrical conductivity of Ag is 6.301 × 10^7^ S/m, which is more than 63 times that of graphene (1 × 10^6^ S/m) [54,55]. This provides an explanation of why the sheet resistance of the GAG is far less than graphene.

## 4. Conclusions

Based on the study of materials of theindex dielectric layer to cover NPs andthe radius of nanoparticles and period of the plasmon, a graphene-metal nanocomposite transparent conductive layer of GAG was designed.Due to the plasma excited effect of Ag NP and double-layer graphene electromagnetically induced effect, the effective radiated power of the GAG transparent conductive layer is reaching 1.33, its carrier motilities are 14,729 cm^2^/(v s),and its sheet resistances are 16.3 ohm/sq. The PL of LED has significantly improved due to the performance of the GAG transparent conductive layer. In conclusion, GAG is a sounded, promising, transparent conductive layer to meet the future demand of light devices.

## Figures and Tables

**Figure 1 polymers-11-01037-f001:**
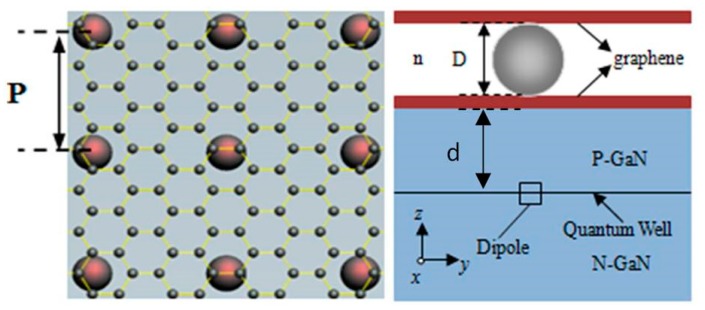
The structure schematic diagrams of graphene-Ag-graphene(GAG).

**Figure 2 polymers-11-01037-f002:**
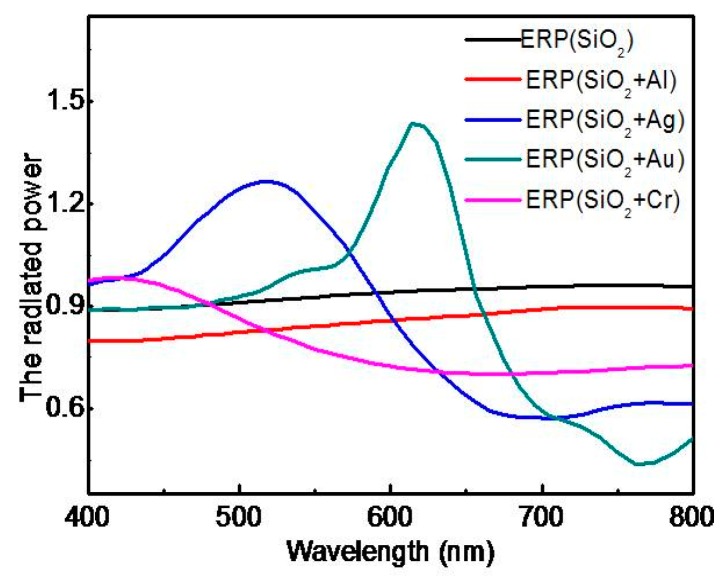
The radiated powers of without metal and with four kinds of metal nanoparticles (NPs).

**Figure 3 polymers-11-01037-f003:**
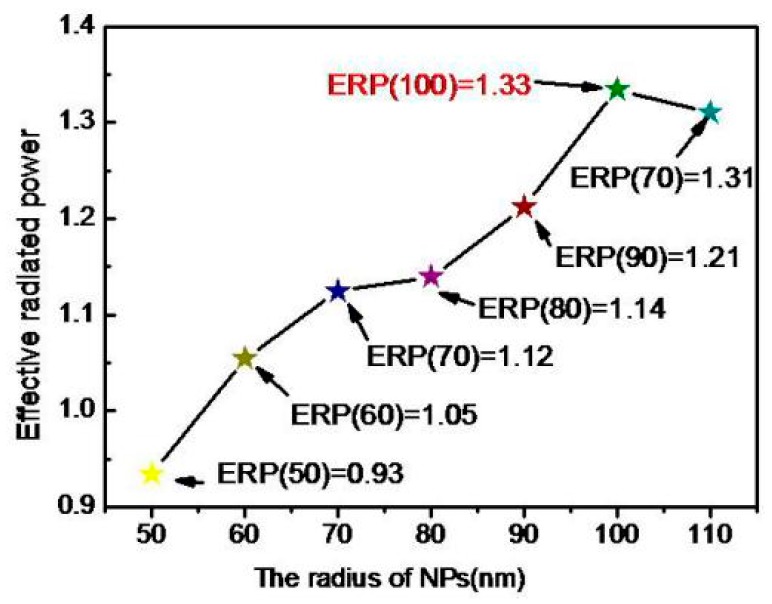
The effective radiated powers of seven kinds of Ag NPs under 600 nm light wave radiation by finite-difference time-domain method (FDTD) simulation.

**Figure 4 polymers-11-01037-f004:**
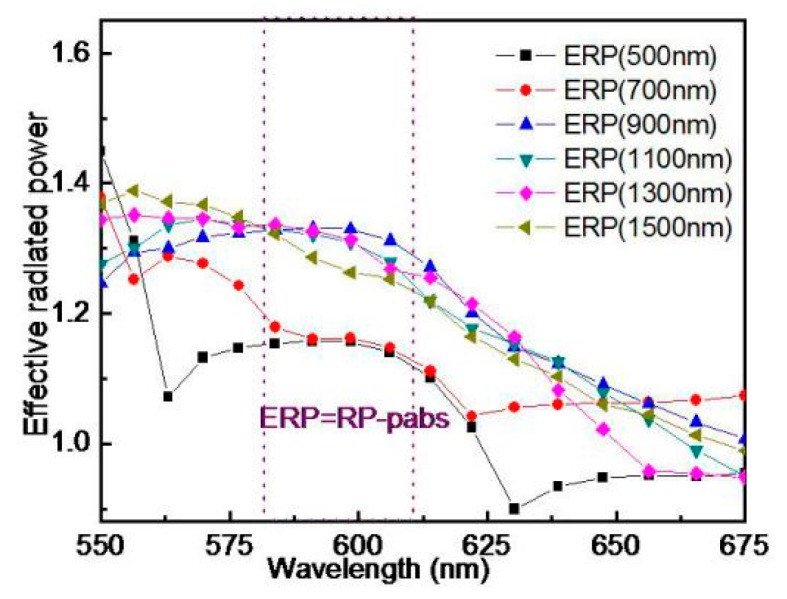
The effective radiated powers of different spacing of the Ag NPs by FDTD simulation.

**Figure 5 polymers-11-01037-f005:**
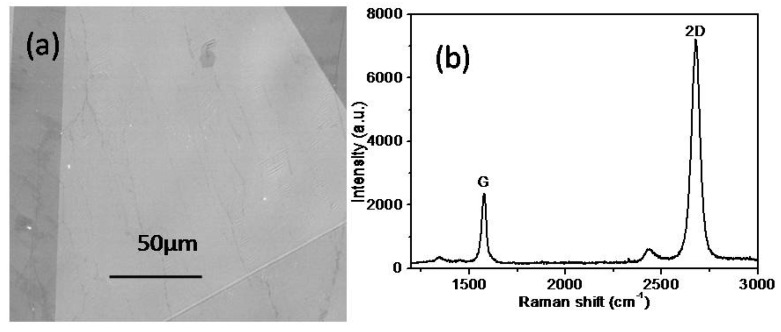
(**a**) SEM image and (**b**) Raman spectrum of the as-prepared graphene.

**Figure 6 polymers-11-01037-f006:**
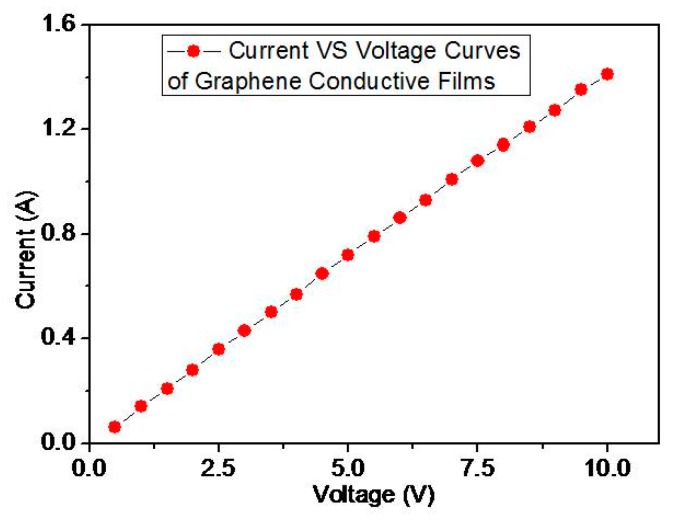
I-V curve of the graphene.

**Figure 7 polymers-11-01037-f007:**
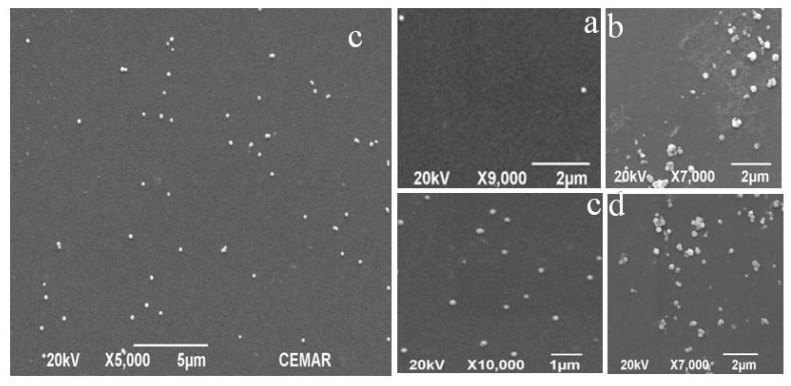
SEM images of four spin-coated samples: (**a**) 1000 r/min 30 s, (**b**) 600 r/min 30 s, (**c**) 600 r/min 10 s and (**d**) 0 r/min.

**Figure 8 polymers-11-01037-f008:**
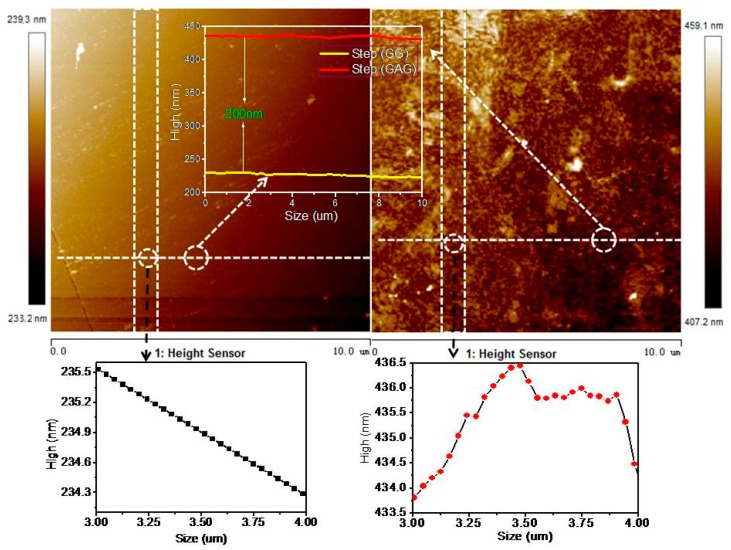
Atomic force microscope (AFM) image of GAG transparent conductive layer and the AFM of graphene-graphene.

**Figure 9 polymers-11-01037-f009:**
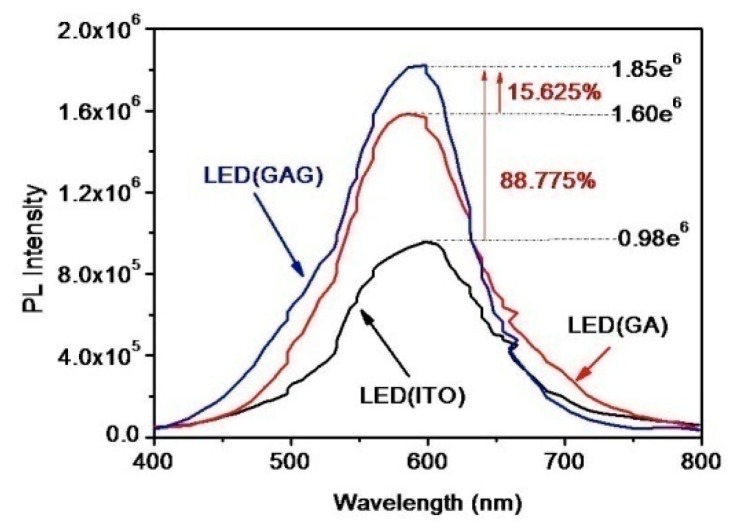
The photoluminescence (PL) spectra of light-emitting diodes (LED) with indium tin oxide (ITO), with graphene-Ag (GA) and with GAG.

**Figure 10 polymers-11-01037-f010:**
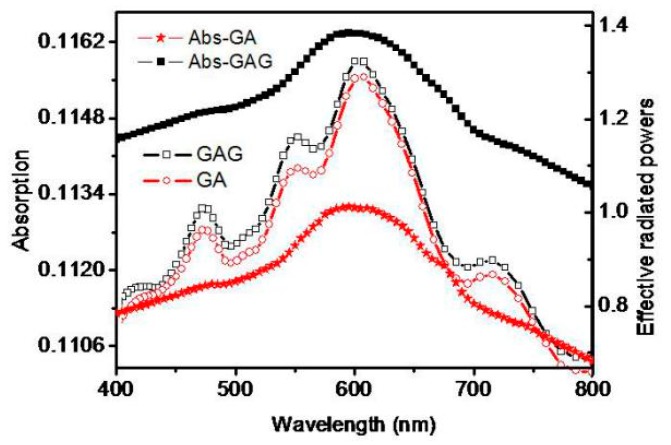
The absorption spectrum and the effective radiated powers of GA and GAG.

**Figure 11 polymers-11-01037-f011:**
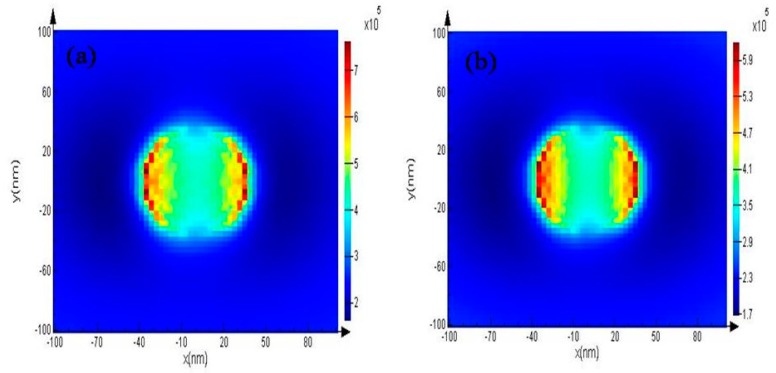
(**a**) current induced on GAG by 600 nm light wave by FDTD simulation; (**b**) The current induced on GA by 600 nm light wave by FDTD simulation.

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
