# Peer review of "Fabrication of Graphene-Metal Transparent Conductive Nanocomposite Layers for Photoluminescence Enhancement"

_polymers, 2019, doi:10.3390/polym11061037_

Round 1
Reviewer 1 Report
Fabrication of Graphene-metal Transparent Conductive nanocomposite Layer for Photoluminescence Enhancement
Recommendation: major revision
Summary: In this manuscript, Huang et. al. studied fabrication of graphene-silver-graphene (GAG) nanocomposite layer. It was found that the square resistance and electron resistance of the GAG is better than the single graphene layer. There are sever major issues to be solved, and I would recommend a major revision for this manuscript.
Major issues
1. The title of the manuscript indicates the GAG layer enhances photoluminescence, however, there is no experimental results that support the enhancement of photoluminescence.
2. Introduction doesn’t provide sufficient information about the background of the field and the work.
3. There is lack of experiment method.
a. Page 3, line 97, the thickness of the graphene was set “0.34 nm”. Please provide experimental evidence about the graphene thickness.
b. Page 4, line 155, for the equation to determine the ERP, please provide how the Pabs is determine.
c. For FDTD simulation, please provide details of how the FDTD simulation was set up.
d. Page 5, line 170, please provide how the size of the six period was determined.
e. Page 7, the fluctuation of the AFM of the GAG layers was explained by the Ag nanoparticles, it is recommended to provide the AFM of graphene-graphene layers as comparison.
f. Page 7, line 235, the thickness of the GAG layer was set 200.68 nm, please provide how the thickness was determined.
4. Page 3, Figure 1, please provide what P-GaN means. In addition, no ITO and P-GaN were used in the GAG sample preparation, why do the authors have ITO and P-GaN were used in Figure 1?
5. Please check the acronyms, and provide the full names of the acronyms when they were firstly used in the manuscript.
Author Response
Dear Reviewers,
Thank you very much for your constructive comments and suggestions concerning our manuscript entitled “Fabrication of Graphene-metal Transparent Conductive Nanocomposite Layer for Photoluminescence Enhancement” (Manuscript ID: polymers-509061). The comments are very helpful for us to improve the quality of our work. We have studied your comments carefully and substantially revised our manuscript hoping to meet with the high standard for publication on Polymers. The revisions are addressed point by point below and the realted changes are highlighted in the revised manuscript.
Thank you for your meticulous and patient reviews and guidance. I hope my adjustment can match your modification opinion. If there are still some inappropriate places, please point out and give us the opportunity to revise, so that the research paper will be more perfect. We are looking forward to your good news.
Thank you and best regards.
Yours sincerely,
Hongyong Huang
To Reviewer #1:
Comment 1: The title of the manuscript indicates the GAG layer enhances photoluminescence, however, there is no experimental results that support the enhancement of photoluminescence.
Reply: Thanks for your very thoughtful suggestion. The experimental results that support the enhancement of photoluminescence are given in Page 8 Line 8.“ITO, GAG and GA were used as transparent conductive layer for the same type LED, their PL spectra shown in Fig. 9 indicate that GA and GAG are more suitable for transparent conductive layers than ITO. The PL peak value of GAG is 15.625% higher than that of GA.” Page 8 Line 8.
Comment 2: Introduction doesn’t provide sufficient information about the background of the field and the work.
Reply: Thanks for your very thoughtful suggestion. The introduction is inadequate in content and lacks in logic. So the introduction and relevant references, especially on PL, have been added.
Comment 3: Page 3, line 97, the thickness of the graphene was set “0.34 nm”. Please provide experimental evidence about the graphene thickness.
Reply: Thank you for your comment. 0.34 nm is the thickness of monolayer graphene, which is supported by reference papers. This is also the graphene thickness parameter set by FDTD software simulation in this paper.
Comment 4: Page 4, line 155, for the equation to determine the ERP, please provide how the Pabs is determined.
Reply: Thanks for your very thoughtful suggestion. The absorbed power of the NP (Pabs) determined by its size and material. Page 4 Line 7, “…another power monitor surrounding the Ag NP is used to measure the absorbed power of the NP (Pabs) determined by its size and material.”
Comment 5: For FDTD simulation, please provide details of how the FDTD simulation was set up.
Reply: Thanks for your very thoughtful suggestion. “2.6 Simulation method” is rewritten, which is about the details of how the FDTD simulation was set up.
Comment 6: Page 5, line 170, please provide how the size of the six period was determined.
Reply: Thanks for your very thoughtful suggestion. In fact, we do more than 20 kinds of periods which are set by references. Page 6 Line 8.The six periods which are a regular set of periods selected from more than a dozen research periods are 500 nm, 700 nm, 900 nm, 1100 nm, 1300 nm, and 1500 nm, respectively, and their ERP are shown in Fig. 4.
Comment 7: Page 7, the fluctuation of the AFM of the GAG layers was explained by the Ag nanoparticles, it is recommended to provide the AFM of graphene-graphene layers as comparison.
Reply: Thanks for your very thoughtful suggestion. Your comment makes the structure of GAG clearer. So we provide the AFM of graphene-graphene layers as comparison in Fig. 8.
Comment 8: Page 7, line 235, the thickness of the GAG layer was set 200.68 nm, please provide how the thickness was determined.
Reply: Thank you for your comment. The specific size was deleted, but it was also mentioned that the thickness of monolayer graphene was 0.34 nm, the diameter of Ag was 200 nm, so the GAG was 200.68 nm.
Comment 9: Page 3, Figure 1, please provide what P-GaN means. In addition, no ITO and P-GaN were used in the GAG sample preparation, why do the authors have ITO and P-GaN were used in Figure 1?
Reply: Thanks for your very thoughtful suggestion. The Fig.1 was redrawn and ITO was deleted because there was no ITO in the simulation.
Comment 10: Please check the acronyms, and provide the full names of the acronyms when they were firstly used in the manuscript.
Reply: Thanks for your very thoughtful suggestion. There are some errors in this detail in some paragraphs. It has been adjusted.

Reviewer 2 Report
This article is dealt with the optimization of nanocomposite layer consisting of graphene and Ag for photoluminescence enhancement. To prove the enhancement of GAG when it comes to photoluminescence, several measurements, such as Raman spectra, SEM, and AFM, and simulations were performed. Despite their endeavors, there are critical questions and ambiguities. For instance, one of the things is that the title, ‘Fabrication of Graphene-metal Transparent Conductive nanocomposite Layer for Photoluminescence Enhancement’ seems not to be met to the contents and explanations in the article. To be specific, although each component was measured to verify their enhancement, the investigation of the nanocomposite is insufficient and is not based on a valid conclusion. Except for it, many things are obscure and representative questions are as follows.
Abbreviations should be checked. (e.g. ‘MQWs’ is no justified in the section of Introduction.
CVD graphene was subjected through a plasma cleaning technique, and then the ability of hydrophilicity for the graphene was obtained. Generally, it is known that the functional groups affect the electrical properties of it; thus, it would be better for the authors to adduce the electrical properties of the modified graphene.
Some parameters are not notified, especially in Drude model. Moreover, it should be changed from ‘Drude mode’ to ‘Drude model’.
Almost simulations was conducted without detailed information.
Please, show the calculation of ‘Drude model’. There are not propose the calculation process of author.
In figure7, the SEM image showed spin-coated samples. To confirm the uniform of GA surface, need to show the SEM image of more equal magnification.
Author Response
Dear Reviewers,
Thank you very much for your constructive comments and suggestions concerning our manuscript entitled “Fabrication of Graphene-metal Transparent Conductive Nanocomposite Layer for Photoluminescence Enhancement” (Manuscript ID: polymers-509061). The comments are very helpful for us to improve the quality of our work. We have studied your comments carefully and substantially revised our manuscript hoping to meet with the high standard for publication on Polymers. The revisions are addressed point by point below and the realted changes are highlighted in the revised manuscript.
Thank you for your meticulous and patient reviews and guidance. I hope my adjustment can match your modification opinion. If there are still some inappropriate places, please point out and give us the opportunity to revise, so that the research paper will be more perfect. We are looking forward to your good news.
Thank you and best regards.
Yours sincerely,
Zhiyou Guo
To Reviewer #2:
Comment 1: ‘Fabrication of Graphene-metal Transparent Conductive nanocomposite Layer for Photoluminescence Enhancement’ seems not to be met to the contents and explanations in the article. To be specific, although each component was measured to verify their enhancement, the investigation of the nanocomposite is insufficient and is not based on a valid conclusion.
Reply: Thanks for your very thoughtful suggestion. The experimental results that support the enhancement of photoluminescence are given in Page 8 Line 8.“ITO, GAG and GA were used as transparent conductive layer for the same type LED, their PL spectra shown in Fig. 9 indicate that GA and GAG are more suitable for transparent conductive layers than ITO. The PL peak value of GAG is 15.625% higher than that of GA.” Page 8 Line 8.
Comment 2: Abbreviations should be checked. (e.g. ‘MQWs’ is no justified in the section of Introduction.
Reply: Thanks for your very thoughtful suggestion. I have checked and change. If there are such questions in the article, welcome to point out again.
Comment 3: CVD graphene was subjected through a plasma cleaning technique, and then the ability of hydrophilicity for the graphene was obtained. Generally, it is known that the functional groups affect the electrical properties of it; thus, it would be better for the authors to adduce the electrical properties of the modified graphene.
Reply: Thanks for your very thoughtful suggestion. Because the experimental preparation is realistic, but thank you for your valuable comments, I will consider your suggestions in the follow-up study.
Comment 4: Some parameters are not notified, especially in Drude model.
Reply: Thanks for your comment. I have checked and change. If there are such questions in the article, welcome to point out again.
Comment 5: Moreover, it should be changed from ‘Drude mode’ to ‘Drude model’. Almost simulations were conducted without detailed information. Please, show the calculation of ‘Drude model’.
Reply: Thanks for your comment. I have checked and change. If there are such questions in the article, welcome to point out again.
Comment 6: There are not propose the calculation process of author.
Reply: Thanks for your very thoughtful suggestion. “2.6 Simulation method” is rewritten, which is about the details of how the FDTD simulation was set up.
Comment 7: In figure7, the SEM image showed spin-coated samples. To confirm the uniform of GA surface, need to show the SEM image of more equal magnification.
Reply: Thanks for your very thoughtful suggestion. We add the sample C`s SEM image of more equal magnification to show that it is more uniform in a wider range.

Round 2
Reviewer 1 Report
Recommendation: minor revision
Summary: with the revision, the publication is worth for publication at Polymers with some minor issues addressed.
Minor issues:
1. Page 8, Figure 9, the authors claimed the fluctuation of GAG layers are due to the presence of Ag NPs. However, in Figure 9, it is difficult to check the fluctuation of GAS layers and the difference between GAG and GG layers. It is recommended to zoom in the AFM of GAG and GG layer to check if the GAG and GG layers are difficult in fluctuation and if the fluctuation of GAG layer is due to the Ag NPs.
Author Response
Response to Reviewers
Dear Reviewers,
Thank you very much for your constructive comments and suggestions concerning our manuscript entitled “Fabrication of Graphene-metal Transparent Conductive Nanocomposite Layer for Photoluminescence Enhancement” (Manuscript ID: polymers-509061). The comments are very helpful for us to improve the quality of our work. We have studied your comments carefully and substantially revised our manuscript hoping to meet with the high standard for publication on Polymers. The revisions are addressed point by point below and the related changes are highlighted in the revised manuscript.
To Reviewer 1:
Comment 1:Page 8, Figure 9, the authors claimed the fluctuation of GAG layers are due to the presence of Ag NPs. However, in Figure 9, it is difficult to check the fluctuation of GAS layers and the difference between GAG and GG layers. It is recommended to zoom in the AFM of GAG and GG layer to check if the GAG and GG layers are difficult in fluctuation and if the fluctuation of GAG layer is due to the Ag NPs.
Reply: Thanks for your thoughtful suggestion. We thought Figure 9 you mentioned should be Figure 8, which are the AFM images of GAG transparent conductive layer and graphene-graphene layers (GG). Accordingly, zoomed AFM images of GAG and GG layers indicated the different fluctuations. The high profile of GAG reflected the distribution of silver particles. The fluctuation of GG is very gentle, showing the characteristics of high quality graphene. Please check the revised Fig. 8 in Page 8 of the main text.
Once again, thank you for your comments and suggestions.
Yours Sincerely,
Prof. Zhiyou GUO

Reviewer 2 Report
This manuscript demonstrated the Fabrication of nanocomposite layer consisting of graphene and Ag for photoluminescence enhancement. The authors revised the manuscript in accordance with comments.
This manuscript is well progressed, and sufficiently modified. Adjusted ‘Drude model’ and calculation process of author also helps to clarify how to the surface conductivity can be retrieved.
For these reasons, I think this manuscript can be accepted for publication in Polymers.
Author Response
Response to Reviewers
Dear Reviewers,
Thank you very much for your constructive comments and suggestions concerning our manuscript entitled “Fabrication of Graphene-metal Transparent Conductive Nanocomposite Layer for Photoluminescence Enhancement” (Manuscript ID: polymers-509061). The comments are very helpful for us to improve the quality of our work.
Once again, thank you for your comments and suggestions.
Yours Sincerely,
Prof. Zhiyou GUO